# Functionality of Yeast β-Glucan Recovered from *Kluyveromyces marxianus* by Alkaline and Enzymatic Processes

**DOI:** 10.3390/polym14081582

**Published:** 2022-04-13

**Authors:** Pilanee Vaithanomsat, Nutthamon Boonlum, Chanaporn Trakunjae, Waraporn Apiwatanapiwat, Phornphimon Janchai, Antika Boondaeng, Kanokwan Phalinphattharakit, Hataitip Nimitkeatkai, Amnat Jarerat

**Affiliations:** 1Kasetsart Agricultural and Agro-Industrial Product Improvement Institute (KAPI), Kasetsart University, Chatuchak, Bangkok 10900, Thailand; aappln@ku.ac.th (P.V.); mmooolyy.2539@gmail.com (N.B.); aapcpt@ku.ac.th (C.T.); aapwpa@ku.ac.th (W.A.); aappmj@ku.ac.th (P.J.); aapakb@ku.ac.th (A.B.); 2Food Technology Program, Mahidol University, Kanchanaburi Campus, Saiyok, Kanchanaburi 71150, Thailand; kanokwan.nia@student.mahidol.ac.th; 3School of Agriculture and Natural Resources, University of Phayao, Muang, Phayao 56000, Thailand; hataitip.ni@up.ac.th

**Keywords:** alkaline recovery, enzymatic recovery, functionality, β-glucan, yeast

## Abstract

β-Glucan (BG), one of the most abundant polysaccharides containing glucose monomers linked by β-glycosidic linkages, is prevalent in yeast biomass that needs to be recovered to obtain this valuable polymer. This study aimed to apply alkaline and enzymatic processes for the recovery of BG from the yeast strain *Kluyveromyces marxianus* TISTR 5925. For this purpose, the yeast was cultivated to produce the maximum yield of raw material (yeast cells). The effective recovery of BG was then established using either an alkaline or an enzymatic process. BG recovery of 35.45% was obtained by using 1 M NaOH at 90 °C for 1 h, and of 81.15% from 1% (*w*/*v*) hydrolytic protease enzyme at 55 °C for 5 h. However, BG recovered by the alkaline process was purer than that obtained by the enzymatic process. Fourier transform infrared (FTIR) and nuclear magnetic resonance (NMR) spectroscopy confirmed the purity, the functional groups, and the linkages of BG obtained from different recovery systems and different raw materials. The results of this study suggest that an alkaline process could be an effective approach for the solubilization and recovery of considerable purity of BG from the yeast cells. In addition, the obtained BG had comparable functional properties with commercially available BG. This study reveals the effectiveness of both chemical and biological recovery of BG obtained from yeast as a potential polymeric material.

## 1. Introduction

β-Glucan (BG) is one of the most abundant forms of polysaccharides that consists of heterogeneous groups of glucose polymers connected by β-1,3/β-1,6-glycosidic linkages. BG has various functional properties that can be applied in the food industry for the preparation of soups, sauces, beverages, and other food products in which BG acts as an emulsifier, stabilizer, and thickening agent [1]. BG is also recognized for its various health benefits, including its potential to diagnose diseases and disorders including cancer [2], nutraceutical, anti-inflammatory, and immune modulatory effects [3], and effects on blood glucose levels and insulin resistance [4], blood lipid levels [5], and gut microflora [6].

BG comprises a significant amount of biomass and is essential for cell function. It is commonly found in the cell walls of yeasts, bacteria, fungi, algae, and grains. Continued study in this area has led the identification of the most efficient techniques to obtain glucanic compounds with various biological features, which are influenced by the length of the polymer chain, the degree of solubility, and the purity of the extract. A number of extraction techniques have been considered to recover potent BG from microbial and plant sources. The potential application and the process of obtaining BG from spent brewer’s yeast have been highlighted in a recent review [7].

We previously reported the production of ethanol by a yeast newly isolated from fruit sources by our laboratory, which was identified as an industrial thermotolerant yeast strain of *Kluyveromyces marxianus* TISTR 5925 [8]. After alcohol fermentation, the spent yeast cells, which are separated as a by-product, can be used as a source of BG. Previous studies revealed that BG exhibited antioxidant activity [9] as well as other bioactive properties [10]. These yeasts contain many valuable and bioactive compounds, which has stimulated much interest in their utilization. There is ongoing interest in the BG sources with varying properties and applications that require research.

Despite the fact that there have been consistent studies on BG of *Saccharomyces cerevisiae*, to the best of our knowledge, there are few studies on the BG of yeast other than *S. cerevisiae* such as *Kluyveromyces marxianus*. Therefore, we report the alkaline and enzymatic recovery processes of BG accumulated from potential sources of *K. marxianus*. The obtained BG samples were characterized using Fourier transform infrared (FTIR) and nuclear magnetic resonance (NMR) spectroscopy. The functionality of BG, including water-holding capacity, water-binding capacity, swelling capacity, oil-holding capacity, and glucose adsorption capacity, was elucidated for its application as a potential polymeric material.

## 2. Materials and Methods

### 2.1. Yeast Strains

Brewer’s yeast *S. cerevisiae* Kyokai NO. 9 TISTR 5169 was purchased from the Thailand Institute of Scientific and Technological Research (TISTR, Pathum Thani, Thailand). The yeast strain *Kluyveromyces marxianus* TISTR 5925 was isolated from rotten fruit by our laboratory, which was previously identified as an industrial thermotolerant yeast strain [8]. The yeast strains were cultured on YPD agar plates (Difco, Thermo Fisher Scientific Inc., Waltham, MA, USA) and stored at 4 °C for further study.

### 2.2. Biomass Cultivation

The yeast strain of *K. marxianus* TISTR 5925 was streaked on yeast extract peptone dextrose agar (YPD agar; 10 g of yeast extract, 20 g of peptone, 20 g of glucose, 15 g of agar, and 1000 mL of distilled water) and incubated at room temperature (28 ± 2 °C) for 48 h. A single yeast colony was then inoculated into YPD broth (10 g of yeast extract, 20 g of peptone, 20 g of glucose, and 1000 mL of distilled water) and incubated at room temperature for 24 h to obtain a yeast stock solution. A 10% yeast inoculum was further inoculated into YPD broth (10 g of yeast extract, 20 g of peptone, 100 g of glucose, and 1000 mL of distilled water) and incubated at room temperature for 72 h to obtain yeast cells, which were then subjected to further BG extraction. The yeast strain of *S. cerevisiae* Kyokai NO. 9 was treated in the same way as *K. marxianus* TISTR 5925.

Pilot-scale production of the culture was performed in a 10 L bioreactor (MDFT1000, B.E. Marubishi Co., Ltd., Tokyo, Japan) with a 7 L working volume. Yeasts were individual cultured in YPD medium at 30 °C and pH 5.0 for 48 h. The YPD medium was inoculated with an initial cell concentration of approximately 10^7^ CFU/mL (350 mL). An agitation rate of 200 rpm and an aeration rate of 1.0 vvm were used.

The growth of yeasts was measured spectroscopically at 600 nm. The dry weight was measured using 2 mL of culture broth, which was previously diluted to 50 mL, filtered through a pre-dried filter (0.45 μm), and washed twice with 50 mL of normal saline solution. The filtered cells were dried at 100 °C in a hot air oven until a constant weight was obtained and were subsequently cooled at room temperature for 2 h before weighing. The dried cell weight was recorded and expressed as cell mass (g/L).

### 2.3. Recovery of β-Glucan (BG) from Yeast Biomass

#### 2.3.1. Alkaline Recovery of BG

The alkaline recovery process of BG from yeast biomass was slightly modified according to Suphantharika et al. [11]. The yeast pellet from the aforementioned broth culture was collected via centrifugation. To obtain a cell content of 15% (*w*/*v*), distilled water was added and incubated at 55 °C for 24 h with shaking at 120 rpm. After that, the cells were heated at 80 °C for 15 min, then cooled down to room temperature, and harvested via centrifugation. Five volumes of sodium hydroxide solution at varied concentrations (0.5, 1.0, 1.5, 2.0 N) were added, and the mixture was incubated at 90 °C for 1 h with stirring. The mixture was cooled down to room temperature, separated by centrifugation, and washed with water. The BG precipitate was collected by centrifugation and dried at 40 °C under vacuum for 48 h. BG extract was ground into a fine powder using a multi-purpose grinder (BL-T70-PR2, Toshiba, Tokyo, Japan) at medium speed for 2 min, then passed through a stainless steel sieve (mesh No. 40), and stored in aluminum foil bags at 4 °C for further analysis.

#### 2.3.2. Enzymatic Recovery of BG

The enzymatic recovery of BG from yeast biomass was performed according to Vaithanomsat et al. with some modifications [12]. The yeast pellet from the aforementioned broth culture was collected via centrifugation. To obtain a cell content of 15% (*w*/*v*), water was added and incubated at 55 °C for 24 h with shaking at 120 rpm. After that, the cells were heated at 80 °C for 15 min, then cooled down to room temperature, and harvested via centrifugation. Protease enzyme (Alcalase^®^ 2.4 L FG, Novozymes A/S, Kalundborg, Denmark) with a declared activity of 2.4 AU/mg was added at concentrations of 0.25, 0.5, 0.75, 1.0% (*w*/*v*), and the mixture was incubated at 55 °C for 5 h. The cells were then separated by centrifugation, washed with water, and the precipitate was separated by centrifugation. Ethanol was added at a ratio of 1:4 (precipitate:ethanol). The BG precipitate was collected by centrifugation and dried at 40 °C under vacuum for 48 h. BG extract was ground into fine powder using a multi-purpose grinder (BL-T70-PR2, Toshiba, Tokyo, Japan), then passed through a stainless steel sieve (mesh No. 40), and stored in aluminum foil bags at 4 °C for further analysis.

### 2.4. Determination of BG Content

The extracts from alkaline and enzymatic recoveries were quantitatively analyzed for BG content according to the manufacturer’s instructions (Catalog Number K-EBHLG, Megazyme International Ireland Ltd., Bray, Ireland). Briefly, approximately 20 mg of dry sample was mixed with 0.4 mL of 2 M KOH and the mixture was stirred for 30 min in an ice water bath. Then, 1.6 mL of 1.2 M sodium acetate buffer (pH 3.8) was added and mixed well. The 40 µL of Glucamix enzyme mixture was added, mixed in the ice water bath for 2 min and then heated to 40 °C for 16 h. After that, 10 mL of distilled water was added and a 0.1 mL aliquot was reacted with 4 mL of GOPOD (glucose oxidase plus peroxidase and 4-aminoantipyrine) reagent and incubated at 40 °C for 20 min. The absorbance of all solutions was measured at 510 nm against the reagent blank. The reagent blank consisted of 0.1 mL of sodium acetate buffer (200 mM, pH 5.0) and 4.0 mL GOPOD reagent. The reference yeast (1→3, 1→6) BG and all of the reagents used were available in the Megazyme Enzymatic Yeast Beta-Glucan Assay Kit.

### 2.5. Composition Analysis of BG Extract

The contents of moisture, protein, lipid, ash, fiber, and carbohydrates in BG extracts and commercially available BG were analyzed according to the Association of Official Agricultural Chemists (AOAC) [13].

### 2.6. BG Recovery

This percentage value relates the amount of BG extract to the total BG in the yeast cell (dry weight), indicating how effective the extraction process is. The BG content was quantitatively analyzed using a Megazyme Enzymatic Yeast Beta-Glucan Assay Kit as described previously.
$$\%\;\mathrm{BG}\;\mathrm{recovery}=\frac{\mathrm{BG}\;\mathrm{extract}}{\mathrm{weight}\;\mathrm{of}\;\mathrm{BG}\;\mathrm{in}\;\mathrm{yeast}\;\mathrm{cell}\;(g)}\times 100$$

### 2.7. Analysis of Residual Protein

The residual protein in the aqueous solution after the recovery of BG was analyzed. After the BG precipitate was collected, the remaining supernatant was subjected to freeze drying to obtain a solid residue, which was then quantitated for protein content according to the Association of Official Agricultural Chemists (AOAC) [13]. 

### 2.8. Fourier Transform Infrared (FTIR) Spectroscopy

The FTIR analysis of extracted BG was performed using an FTIR spectrometer (Thermo Scientific Nicolet IR200, Waltham, MA, USA). A total of 128 scans were accumulated in attenuated total reflection (ATR) mode with a resolution of 4 cm^−1^. The spectra were obtained in the range from 4000–400 cm^−1^.

### 2.9. Nuclear Magnetic Resonance Spectroscopy (NMR) of BG

The solid-state ^13^C NMR measurements were performed on a Jeol JNM-ECZ-400R/S1 spectrophotometer (JEOL, Ltd., Tokyo, Japan) resonating at 400 MHz. The spectra were obtained at room temperature averaging over 5000–33,000 scans. The chemical shifts were referenced to the TMS using adamantane as an external standard.

### 2.10. In Vitro Functional Properties of BG

#### 2.10.1. Water-Holding Capacity (*WHC*)

*WHC* was determined according to Sangeethapriya and Siddhuraju [14] with some modifications. Briefly, 1 mL of 0.02% sodium azide solution was added to 0.5 g of extracted BG. The mixture was stirred gently and left for 60 min at room temperature. The mixture was then centrifuged at 3000× *g* for 15 min, the supernatant was carefully discarded, and the residue was weighed (*m*). *WHC* was reported as the amount of water per gram of BG (g/g) and calculated as follows:$$\mathrm{WHC}\;(g/g)=\frac{m-0.5}{0.5}$$

#### 2.10.2. Water-Binding Capacity (*WBC*)

*WBC* was determined according to Daou and Zhang [15] with some modifications. Briefly, 15 mL of 0.02% sodium azide solution was added to 0.5 g of extracted BG. The mixture was stirred and left to stand for 18 h at room temperature. After this time, the sample was centrifuged at 3000× *g* for 20 min. The supernatant was carefully discarded, the residue was weighed, and sample was vacuum dried for 2 h at 60 °C. The dried weight was recorded and used for calculation as follows:$$\mathrm{WBC}(g/g)=\frac{\mathrm{residue}\;\mathrm{weight}\;\mathrm{after}\;\mathrm{centrifugation}-\mathrm{residue}\;\mathrm{dry}\;\mathrm{weight}}{\mathrm{residue}\;\mathrm{dry}\;\mathrm{weight}}$$

#### 2.10.3. Swelling Capacity (*SC*)

*SC* was determined according to Daou and Zhang [15]. First, 10 mL of 0.02% sodium azide solution was added to 0.2 g of BG and the mixture was stirred and left at room temperature for 18 h. The bed volume was then recorded. The *SC* was calculated as follows:$$SC\;(\mathrm{mL}/g)=\frac{V_{1}-V_{0}}{W_{0}}$$
where $V_{1}$ is the volume of the hydrated BG, $V_{0}$ is the volume of BG prior to hydration, and $W_{0}$ is the weight of BG prior to hydration.

#### 2.10.4. Oil-Holding Capacity (*OHC*)

*OHC* was determined according to Sangeethapriya and Siddhuraju [14]. Briefly, 10 mL of refined sunflower oil was added to 0.5 g of extracted BG. The mixture was stirred carefully and left for 60 min at room temperature. After being centrifuged at 3000× *g* for 15 min, the supernatant was carefully discarded, and the residue was weighed (*m_r_*). *OHC* was reported as the amount of oil per gram of BG (g/g) and was calculated as follows:$$\mathrm{OHC}\;(g/g)=\frac{m_{r}-m_{d}}{m_{d}}$$
where $m_{r}$ is the residue weight, which contained the oil (g), and $m_{d}$ is the original weight of BG (g).

#### 2.10.5. Glucose Adsorption Capacity (*GAC*)

*GAC* was determined according to Peerajit et al. [16] with some modifications. The extracted BG of 0.5 g was mixed with glucose solution (20–50 mmol/L) and incubated at 37 °C for 6 h. After that, the sample was centrifuged at 4000× *g* for 20 min and the supernatant was subjected to glucose determination using DNS colorimetric method. The adsorbed glucose was calculated as the amount of glucose retained by the sample (mmol/g).
$$GAC\left(g/g\right)=\frac{\left(C_{i}-C_{s}\right)\times V_{i}}{W_{s}}$$
where $C_{i}$ is the glucose concentration of the original solution (mmol/L), $C_{s}$ is the supernatant glucose content when adsorption reached equilibrium (mmol/L), $W_{s}$ is the weight of BG (g), and $V_{i}$ is the supernatant volume (mL).

### 2.11. Statistical Analysis

All experimental data were from the calculation of at least 9 replicates and reported as mean ± SD. Analysis of variance (ANOVA) was performed by Duncan’s multiple-range test (DMRT) using SPSS software (SPSS for Windows, SPSS Inc., Chicago, IL, USA) at *p* ≤ 0.05.

## 3. Results and Discussion

### 3.1. Biomass Cultivation

Figure 1 shows the production profile of the yeasts *K. marxianus* TISTR 5925 and *S. cerevisiae* Kyokai NO. 9 in the 10 L bioreactor. Both yeasts similarly took around 6 h to adapt to the environment (during the lag phase), then reached the highest growth at 12–18 h before going through a stationary phase at 24 h. In our study, yeast cells (about 3 g/L) were harvested after 48 h of cultivation, in which they contained initial BG at 12–15 g from 100 g of *S. cerevisiae* Kyokai NO. 9 biomass and 8–10 g from 100 g of *K. marxianus* TISTR 5925 biomass. The industrial thermotolerant yeast strain of *K. marxianus* TISTR 5925 grew and provided a relative higher cell mass than *S. cerevisiae* Kyokai NO. 9, indicating its applicability as a source of BG. It is a competitive strain that has demonstrated excellent ethanol production and cell growth efficiency [8].

The obtained results of this study showed comparable cell mass to other previous reports [8,17]. It has been reported that cell mass and the BG production (8–12% *w*/*w*) were maximized after 48 h [17]. Avramia and Amariei [7] reported that the total carbohydrate content of brewer’s yeast cells (*S. cerevisiae*) usually varied by more than 50% (of which 12–21% are BG), depending on growth condition, medium, and cell age. This study yielded similar content of initial BG.

### 3.2. Recovery of BG from Yeast Biomass

In this study, enzymatic analysis using the Megazyme assay kit was used to assess the content of BG in the samples. The obtained results (Table 1) were similar to those reported by other authors who used this method, as reviewed by Avramia and Amariei [7].

Insoluble parts or BG extracts from the alkaline and enzyme recovery processes were examined. It was found that the extract contained BG in the range of 43–44% (Table 1). The lower percentage yield from alkaline recovery process was probably due to the strong base of NaOH, which could degrade polymer chains, thereby resulting in a lower yield of BG extracts [18].

It has been reported that BG usually occurs at low concentration (~5% *w*/*v*) in cereals [19,20]. Microorganisms have a wide range of BG levels. Optimized yields of BG recovered from baker’s yeast (*S. cerevisiae*) were only 5–7%, nevertheless *Euglena* can accumulate BG intracellularly to more than 90% [21]. The quantity of BG in seaweeds is mostly dependent on the species. *Durvillaea antarctica* stipes and holdfasts account for 33% and 5% of the BG, respectively [22]. The content of BG in mushrooms also contributes widely, from about 3.1% to 46.5% [23].

The recovery efficiency improved with increased alkaline concentration, as shown in Figure 2a, and with increased enzyme dose (Figure 2b). The highest BG recovery (81.15%) was observed in *S. cerevisiae* Kyokai NO. 9 using 1% (*w*/*v*) protease enzyme, whereas 35.45% recovery was obtained at an extraction temperature of 90 °C in 1.0 N alkaline solution.

Using 1.0 N NaOH at 90 °C for 1 h, 35.45% BG recovery was relatively low (2.3-fold) when compared to 1% (*w*/*v*) hydrolytic protease enzyme at 55 °C for 5 h. On the other hand, protein impurity of the enzymatic recovery was relatively high (3-fold) when compared to that of the alkaline recovery (Figure 3), which was also confirmed through the FT-IR analysis of the extracted BG. The optimum alkaline conditions were 90 °C for 1 h with 1.0 N NaOH and a sample to alkaline solution ratio of 1:5 (*w*/*v*), while the optimum enzymatic conditions were at 55 °C for 5 h using 1.0% (*w*/*v*) protease enzyme with cell suspension.

As shown in Figure 2a, the low BG recovery percentage from the alkaline process was probably due to the strong base, which is known to degrade polymer chains and thus result in a lower BG yield (Table 1). Liu et al. [18] also reported that the reduced yield of BG (8.4 −10.4%) obtained after alkaline recovery.

Figure 3 shows protein content left in aqueous solution after the recovery of BG from *K. marxianus* TISTR 5925 and *S. cerevisiae* Kyokai NO. 9 by alkaline and enzymatic methods under their optimum conditions. These were consistent with protein contents detected in extracted BG (Table 2), indicating that an alkaline solution was more efficient for the removal of protein from BG when compared with an enzymatic solution. This may be due to the different protein removal mechanisms of alkaline and enzymatic solutions. The cellular proteins, nucleic acids, mannans, and polar lipids can be hydrolyzed and solubilized in the supernatant fraction after treatment with hot alkali, leaving insoluble BG in the solids fraction [24]. Conversely, enzymatic recovery using protease (serine endopeptidase) hydrolyzed the internal peptide bonds of the protein into short peptide chains and may not act quantitatively. The alkaline process used in this study was found to be a promising method of separating and purifying BG from yeasts since the three-dimensional structure may not be affected, and if this occurs, drying at 55–60 °C can be used to recover random transitions in the polymer chain [25].

### 3.3. Composition of K. marxianus, S. cerevisiae and Commercially Available BG

The composition of BG extracted from *K. marxianus* TISTR 5925, *S. cerevisiae* Kyokai NO. 9, and commercially available BG is shown in Table 2. The major components of the extracted BG from yeasts were carbohydrates (52.05–67.21%) and fiber (18.58–21.64%), whereas fat (2.98–4.78%), protein (3.79–14.8%), and ash (0.33–1.11%) levels were relatively low.

The water content of BG was 5.62–6.89%. Compared to enzymatic recovery, BG from alkaline recovery had lower protein, fat, and ash contents, but higher carbohydrate content. In total, carbohydrates of BG from *K. marxianus* TISTR 5925 recovered using alkaline processing comprised 67.21% of the extract (Table 2), of which 43.40% was BG (Table 1) and 4.96% of the biomass was proteins.

To reduce the protein content, it has been reported that alkaline treatment solubilized a proportion of the cell wall mannoproteins [26]. From the results, it was found that the extracted BG had a similar protein impurity percentage from 3.90% to 5.52% when using alkaline–acid treatment [27], whereas 4.30% protein with 2.68% lipid was obtained by the combination of hot water and homogenization at high pressure [18].

### 3.4. FTIR Spectra of BG Obtained by Alkaline and Enzymatic Recovery

Fourier transform infrared spectroscopy (FTIR) was used to characterize the structure of BG obtained from yeasts and commercially available BG. The ATR FT-IR spectra (Figure 4) contains three broad absorption bands: one at the 3400 cm^−1^ region that can be assigned to the hydroxyl stretching vibration of the polysaccharide, implying a strong O-H bond; a second band at the region between 1318–1420 cm^−1^, occurred from bending modes of CH2, CH, and OH [28]; and a third broad band that can be attributed to polysaccharides at 1200–800 cm^−1^ corresponding to C-O, C-C stretching, and COH bending modes [28]. The peak between 2820 and 3000 cm^−1^ implies C-H stretching that is attributed to CH groups. The C-O bonds of the alcohol groups are responsible for the main peak at 1035 cm^−1^. The linear structure of β-glucan linked through 1–3 linkages agree with peak shoulder at 1080 cm^−1^ and peak at 1160 cm^−1^ [29]. Additionally, the peak at 887 cm^−1^ was a characteristic band that correlates with the occurrence of β-glycosidic bond, i.e., C-H deformation mode related to β-linkages [30,31,32]. Similar observations have been reported for the yeast cell wall preparation [31]. In Figure 4, the analyzed spectra of the BG extracted by alkaline recovery show differences in the absorption intensity in the region between 1400–1600 cm^−1^. These indicate the appearance of functional groups corresponding to proteins, such as amide bonds and aromatic rings. The presence and differences in protein content in samples were also confirmed by the Kjeldahl method (Table 1 and Table 2). It was noticeable that the BG using alkaline recovery had relatively low protein content than that using enzymatic recovery (Figure 4c,e). The functional groups were revealed by FTIR analysis, which provided a preliminary characterization of BG obtained under both alkaline and enzymatic conditions.

### 3.5. Nuclear Magnetic Resonance Spectroscopy (NMR) of BG

The measured ^13^C solid-state NMR spectra of BG are illustrated in Figure 5. The broad signal at 86.1 ppm relates to (1→3)-linked residues of C3, whereas the peak at 104 ppm corresponds to the C1 carbon in the β-glycosidic bond. The signals of C5, C2, and C4 are superimposed in the range of 65–80 ppm, where signals of (1→6)-linked C3 and C6 carbons are usually detected. In Figure 5a–e, the C6 carbon in the (1→3)- and (1→6)-linkages has signals at 62.3 and 69.3 ppm, respectively [33,34,35,36], but the BG obtained from enzymatic recovery (Figure 5c,e) had shoulder peaks with a relatively large width, especially the BG of *K. marxianus* TISTR 5925 (Figure 5c). The peak position of (1→3)-linked residues of C3 at 86.1 ppm confirms that the β-D-glucan chains are in a triple helix conformation [37], which was found in the commercially available BG and the BG of enzymatic recovery from *K. marxianus* TISTR 5925 (Figure 5a,c). The relatively large width of this peak might be due to the occurrence of different allomorphs in the sample [37,38]. Besides the signals related to the carbons of (1→3, 1→6)-β-D-glucan, additional peaks near 175 ppm (C=O), 56 ppm (C2), and 23 ppm (CH3), which may indicate the presence of chitin, were observed [33,39,40].

It should be noted that a broad peak of the carbonyl group (near 175 ppm) and aliphatic carbon peaks (near 33 ppm) might imply the occurrence of some proteins in the BG obtained from enzymatic recovery of both *K. marxianus* TISTR 5925 and *S. cerevisiae* Kyokai NO. 9 (Figure 5c,e). Hence, the BG obtained by alkaline recovery was a more effective approach for the solubilization and recovery of BG of considerable purity from the yeast cells. From the obtained results, it can be assumed that these signals came from the residual traces of compounds after the isolation of (1→3, 1→6)-β-D-glucan from the yeasts.

### 3.6. In Vitro Functional Properties of BG 

Table 3 shows the in vitro functional properties, including water-holding capacity (*WHC*), water-binding capacity (*WBC*), swelling capacity (*SC*), oil-holding capacity (*OHC*), and glucose adsorption capacity (*GAC*) of *K. marxianus* TISTR 5925, *S. cerevisiae* Kyokai NO. 9, and commercially available BG.

Theoretically, *SC* is the ratio of the volume occupied by BG immersed in an excess of water after equilibration to the actual weight of BG. BG may interact with water through two mechanisms: (i) water trapped in capillary structures due to surface tension strength; and (ii) water retained by hydrogen bonds and dipole formation [41]. The *WHC* reflects the ability of BG to hold water when subjected to external forces, such as centrifugation pressure. The associated water—hydrodynamic water—and physically entrapped water also major contributors to the *WHC* [42]. *OHC* measures the ability of BG to adsorb fat. High *OHC* values correspond to the high ability of BG to prevent fat loss during food processing and reduce serum cholesterol levels by adsorbing fat in the intestinal lumen [43].

As shown in Table 3, there were no significant differences in *WHC* and *SC* among the obtained BG and commercial BG, while the BG of *S. cerevisiae* Kyokai NO. 9 recovered by alkaline processing had higher *WBC* and *OHC* than that recovered by enzymatic processing. The *WBC* values observed in the present study revealed the existence of hydrophilic behavior. Both BG from alkaline processing and commercial BG had a significant *GAC*, which might be attributed to the higher proportion of D-glucopyranose along a chain, which could potentially increase glucose molecule trapping inside the chain network, thereby delaying glucose diffusion. ΒG is a strongly hydrophilic, non-starchy polysaccharide containing numerous hydroxyl groups that enable it to form bonds with reactive groups in other molecules. The functional properties of BG have been reported to be directly related to their origin/source, molecular weight, and structural features [10]. Regarding the BG of *K. marxianus*, the BG of yeast cells cultivated on a lactose-based medium was extracted and characterized; however, no reference was made to the yield of BG and its application [44]. In addition to our results, insoluble BG also has various desirable features. For instance, it goes beyond the gastrointestinal tract without being changed and stimulates specific BG receptors [45]. It has a higher binding affinity to dectin-1, a receptor for β-1,3/β-1,6 chains that stimulates an avalanche of innate and adaptive immune responses [46]. The functional properties of the insoluble BG obtained from *S. cerevisiae* Kyokai NO. 9 using alkaline processing tended to have the same properties commercial BG. Overall, the obtained results of this study showed comparable functional properties to commercially available BG.

## 4. Conclusions

Although BG from various organisms, including oats, barley, bacteria, yeast, mold, algae, and mushrooms, has been studied and characterized, there are still many BG sources with different properties and applications awaiting study. In addition to the conventional yeast strain of *Saccharomyces cerevisiae*, the results of this study revealed that the industrial thermotolerant yeast strain of *Kluyveromyces marxianus* TISTR 5925 was an alternative source of BG, produced in appreciable amounts that may be of commercial value to the ethanol industry as a by-product. Our results suggested that an alkaline system could be an economically feasible process for the solubilization and recovery of considerable purity of BG from the yeast cells. The obtained BG exhibited comparable functional properties to commercially available BG, which can be used as a potential polymeric material.

## Figures and Tables

**Figure 1 polymers-14-01582-f001:**
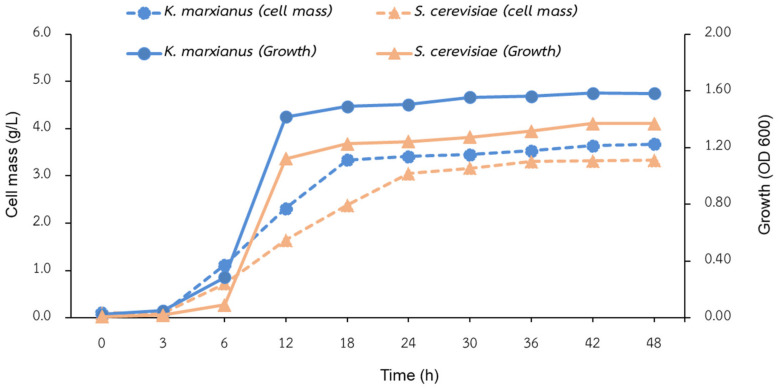
Pilot-scale production profiles of *K. marxianus* TISTR 5925 and *S. cerevisiae* Kyokai NO. 9 in a 10 L bioreactor.

**Figure 2 polymers-14-01582-f002:**
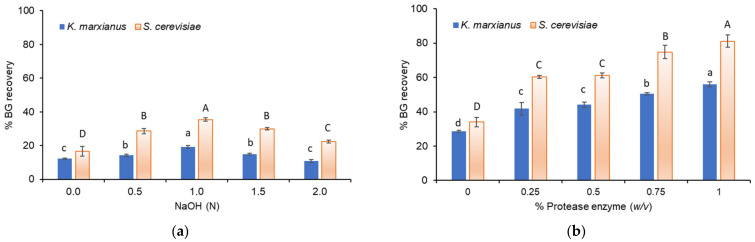
BG recovery by alkaline (**a**) and enzyme (**b**) methods using cells of *K. marxianus* TISTR 5925 and *S. cerevisiae* Kyokai NO. 9 as raw materials. Different letters a–d and A–D indicate statistically significant differences (*p* < 0.05) according to Duncan’s multiple range test. Error bars show mean ± SD (*n* = 3).

**Figure 3 polymers-14-01582-f003:**
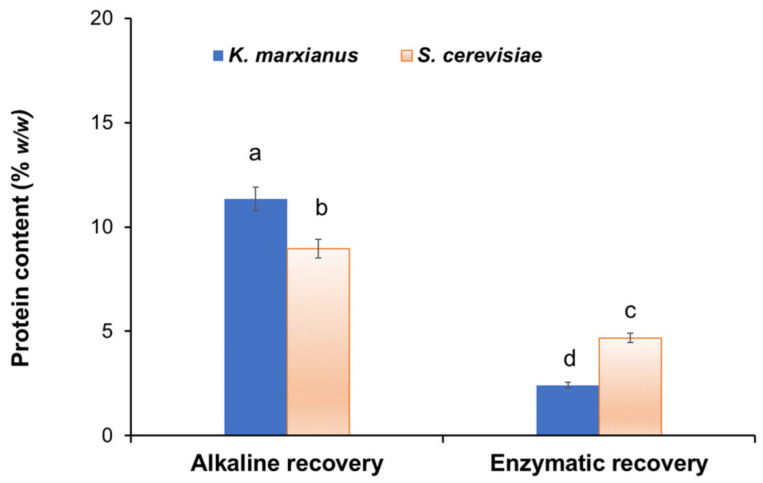
Protein contents in aqueous solution after recovery of BG at the optimum conditions. Different letters a–d indicate statistically significant differences (*p* < 0.05) according to Duncan’s multiple range test. Error bars show mean ± SD (*n* = 3).

**Figure 4 polymers-14-01582-f004:**
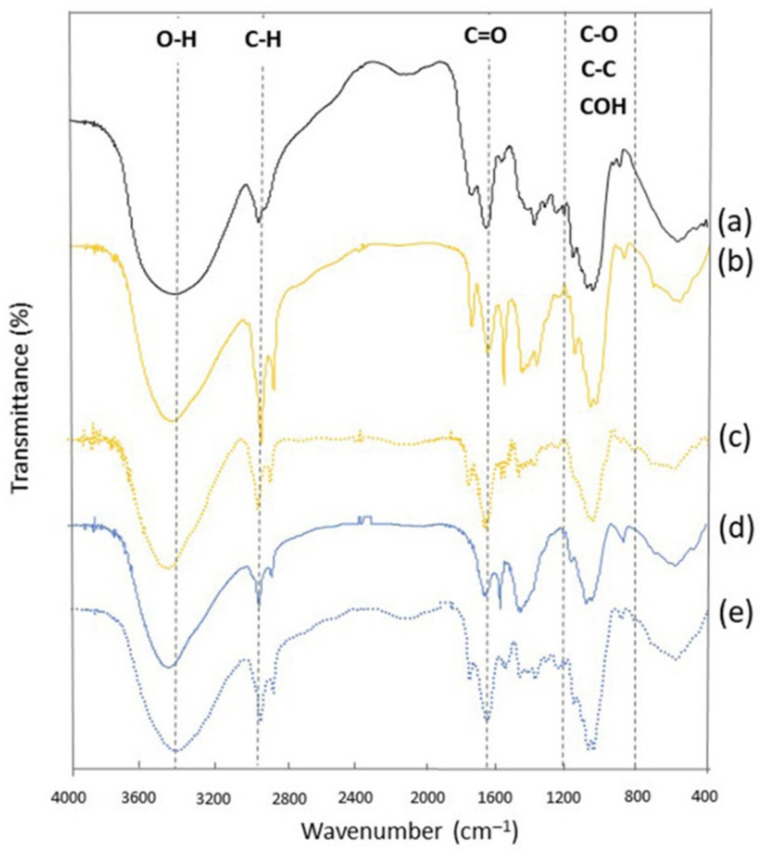
Fourier transform infrared (FTIR) spectra of commercially available BG (**a**), BG obtained from *S. cerevisiae* Kyokai NO. 9 by alkaline (**b**) and enzymatic (**c**) recovery, BG obtained from *K. marxianus* TISTR 5925 by alkaline (**d**) and enzymatic (**e**) recovery.

**Figure 5 polymers-14-01582-f005:**
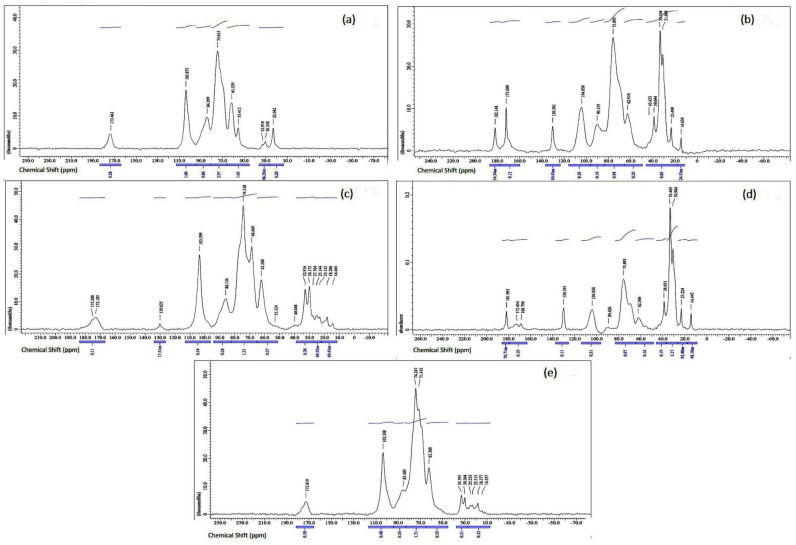
^13^C solid-state NMR spectra of commercially available BG (**a**), BG obtained from *K. marxianus* TISTR 5925 by alkaline (**b**) and enzymatic recovery (**c**), BG obtained from *S. cerevisiae* Kyokai NO. 9 by alkaline (**d**) and enzymatic (**e**) recovery.

**Table 1 polymers-14-01582-t001:** BG content and yield after alkaline and enzymatic recovery from *K. marxianus* TISTR 5925, *S. cerevisiae* Kyokai NO. 9, and commercially available BG.

BG Sources	Original BG in Yeast Biomass (g/100 g)	Recovery Method	Yield (%)	BG in Extract (g/100 g)
*K. marxianus* TISTR 5925	9.05 ± 0.42 ^b^	Alkaline method	6.48 ± 0.58 ^c^	43.40 ± 0.87 ^b^
		Enzymatic method	20.22 ± 0.31 ^b^	44.91 ± 0.60 ^b^
*S. cerevisiae* Kyokai NO. 9	11.87 ± 0.51 ^a^	Alkaline method	8.88 ± 0.39 ^c^	43.55 ± 0.41 ^b^
		Enzymatic method	29.87 ± 0.47 ^a^	43.84 ± 0.79 ^b^
Commercial (*S. cerevisiae*)	-	NA	-	48.69 ± 0.63 ^a^

Mean ± SD values in each column with different small superscript letters a–c are significantly different at *p* < 0.05 level (*n* = 3).

**Table 2 polymers-14-01582-t002:** Composition of BG extracted from *K. marxianus* TISTR 5925, *S. cerevisiae* Kyokai NO. 9, and commercially available BG.

Composition	BG Sources				
	*K. marxianus* TISTR 5925		*S. cerevisiae* Kyokai NO. 9		Commercial (*S. cerevisiae*)
	Alkaline Method	Enzymatic Method	Alkaline Method	Enzymatic Method	
Moisture (%)	5.72 ± 0.01 ^b^	5.62 ± 0.01 ^b^	6.89 ± 0.01 ^a^	6.75 ± 0.01 ^a^	5.30 ± 0.01 ^c^
Carbohydrate (%)	67.21 ± 0.31 ^a^	52.05 ± 0.33 ^e^	65.16 ± 0.11 ^b^	54.59 ± 0.36 ^d^	59.29 ± 0.23 ^c^
Fiber (%)	18.58 ± 0.04 ^e^	21.64 ± 0.09 ^a^	19.93 ± 0.04 ^d^	21.34 ± 0.07 ^b^	20.76 ± 0.02 ^c^
Protein (%)	4.96 ± 0.13 ^d^	14.80 ± 0.25 ^a^	3.79 ± 0.09 ^e^	11.96 ± 0.39 ^b^	11.61 ± 0.12 ^c^
Fat (%)	2.98 ± 0.06 ^d^	4.78 ± 0.03 ^a^	3.91 ± 0.11 ^c^	4.53 ± 0.06 ^b^	2.61 ± 0.01 ^e^
Ash (%)	0.55 ± 0.25 ^bc^	1.11 ± 0.06 ^a^	0.33 ± 0.01 ^c^	0.83 ± 0.06 ^ab^	0.43 ± 0.21 ^c^

Mean ± SD values in a row with different small superscript letters a–e are significantly different at *p* < 0.05 level (*n* = 3).

**Table 3 polymers-14-01582-t003:** Water-holding capacity (*WHC*), water-binding capacity (*WBC*), swelling capacity (*SC*), oil-holding capacity (*OHC*), and glucose adsorption capacity (*GAC*) of *K. marxianus* TISTR 5925, *S. cerevisiae* Kyokai NO. 9, and commercially available BG.

Functional Properties	BG Sources				
	*K. marxianus* TISTR 5925		*S. cerevisiae* Kyokai NO. 9		Commercial (*S. cerevisiae*)
	Alkaline Method	Enzymatic Method	Alkaline Method	Enzymatic Method	
*WHC* (g/g)	1.96 ± 0.01	1.96 ± 0.03	1.98 ± 0.01	1.97 ± 0.01	1.97 ± 0.01
*WBC* (g/g)	0.14 ± 0.01 ^b^	0.18 ± 0.01 ^a^	0.19 ± 0.01 ^a^	0.09 ± 0.01 ^c^	0.13 ± 0.01 ^b^
*SC* (mL/g)	41.50 ± 0.01	41.49 ± 0.14	41.53 ± 0.59	41.72 ± 0.28	41.79 ± 0.27
*OHC* (g/g)	2.80 ± 0.01 ^ab^	2.94 ± 0.20 ^ab^	3.00 ± 0.05 ^a^	2.65 ± 0.05 ^b^	3.00 ± 0.32 ^a^
*GAC* (mmol/g)	0.12 ± 0.01 ^c^	0.17 ± 0.01 ^c^	0.30 ± 0.01 ^b^	0.14 ± 0.01 ^c^	0.39 ± 0.01 ^a^

Mean ± SD values in each row with different small superscript letters a–c are significantly different at *p* < 0.05 level (*n* = 3).

## Data Availability

Data are contained within the article.

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
