# Peer review of "Functionality of Yeast β-Glucan Recovered from Kluyveromyces marxianus by Alkaline and Enzymatic Processes"

_polymers, 2022, doi:10.3390/polym14081582_

Round 1

Reviewer 1 Report

The manuscript shows an interesting topic of the application of alkali and enzymes for β-glucan isolation from biomass of yeast strain of Kluyveromyces marxianus TISTR 5925. It is quite an interesting approach, the authors carried out the examination and comparison of the potential of these two types of yeast biomass treatments for the recovery of β-glucan. The investigation refers only to the specific yeast strain Kluyveromyces marxianus TISTR 5925 newly isolated from 50 fruit sources by authors. The paper is very good in terms of textual structure and scientific content, and it is important since it is based on empirical work.

However, some key parameters in the Materials and methods section must be precisely explained and specified. Formulas must contain appropriate units and should be written in italic. Some figures need to be improved. The suggestions concerning the improvement of these issues, some typographical and spelling errors are provided in the attached file. If authors can make revisions and provide a reasonable explanation based on scientific facts of issues, I think that the work should be published with regard to the systematic way of presentation and discussion of the results.

Reviewer 2 Report

The aim of the study was to isolate beta-glucan polimer from the biomass of new isolate of K. marxianus yeast  using  alkaline and  enzymatic procedure of extraction. Authors used the new-thermotolerant Kluyveromyces yeast isolate as a beta-glucan source. It is noteworthy that the functional properties of the obtained preparation were studied, like water holding capacity, water binding capacity, swelling capacity, oil holding capacity, glucose adsorption capacity. However, considering the obtained biomass yield of the studied yeast in the bioreactor scale after 48 hours of cultivation, the process of beta-glucan production of studied yeast is rather inefficient. At the same time, the only residual information presented on the content of beta-glucan in the biomass of the tested yeast indicates a relatively low content of this polysaccharide, which also negatively affects the efficiency of the process (the amount of preparation obtained). In terms of methodology, the work does not bring new knowledge. Commonly known procedures were used. I also find numerous inaccuracies, both in the literature review and in the methodology of work and inference. I present the most important remarks below. Taking them into account, I do not recommend the work for publication in its current form.

L17: Yeast biomass is rich in …. – not BG

L34-41: Do the indicated properties relate to the glucan of yeast origin? It should be verified because the glucan shows different chemical, physical and biological active properties depending on its origin. It should not be generalized

L133: There is no information how the BG was analysed in yeast cells? The method used to determine the efficiency of glucan extraction requires data on the content of glucan in the biomass. How was the biomass prepared for analysis? L135-137: Have the authors purchased a (1→3, 1→6) β-glucan glucan standard? I am surprised - the indicated enzymatic Megazyme test contains such a standard to control the efficiency of the enzymatic reaction.

L141: „weight of BG in yeast cell” - weigth suggest isolated BG ???

L209-211: The stage of chemical characterization of the preparation should be taken into account beforehand - before determining the functional properties.

L76, 219: The title „Preparation of yeast cells” is unfortunate – it should be „Biomass cultivation”

L224: All results of determination of glucan content in biomass should be indicated (table, graph) - these are fundamental results

L239-263: The methodology first describes isolation under alkaline conditions, then a protease-based process. This sequence should be followed consistently when discussing the results.

At the same time, I note that the % values are relative. The weight of the preparation should be given in relation to biomass used.

L260-263, fig.3 – in methodology there is no information how protein was analysed in aqueous solution after recovery of BG.  This information is also crucial for confirming the correctness of the inference about the effectiveness of the applied methods of glucan isolation and its purification from protein.

L288-289: Authors suggest polimer degradation using NaOH procedure. Can it therefore be recommended for the isolation of glucan of studied yeast? After all, it affects the losses of the process and properties of the preparation.

L291: The title of tabel 1 is non-precise -> the reader must guess what they are meaning BG% and Yield %

L318-323: The statement is incorrect! the obtained preparation did not contain > 80% glucan !!!

L341-432: „Kluyveromyces marxianus TISTR 5925 was also 431 found to be a rich source of BG „ - analyzing the presented results, one cannot conclude in this way

Round 2

Reviewer 2 Report

The authors responded to most of my comments, which significantly improved the work. However, I still identify some shortcomings. I have doubts about the enzymatic method of determining glucan - what exactly test (catalog number) was used to determine beta-glucan? The described procedure (L132-146) does not allow for the determination of beta-glucan, and total glucans. If the authors used the described procedure, the results do not apply to beta-glucan. Moreover, in the described methodology, the acetate buffer should have a pH of 5.0, not a pH of 4.5.